# Diagnostic Method for Enhancing Nitrogen and Phosphorus Removal in Cyclic Activated Sludge Technology (CAST) Process Wastewater Treatment Plant

**Chong Liu [1,\*], Kai Qian [2] and Yuguang Li [1,\*]**

1   101 Institute of the Ministry of Civil Affairs, Environmental Monitoring Center Station of the Ministry of Civil Affairs, Beijing 100070, China
2   Jiangsu Key Laboratory of Anaerobic Biotechnology, School of Environment and Civil Engineering, Jiangnan University, Wuxi 214122, China; qiankai0413@126.com
*   Correspondence: liuchong_101s@163.com (C.L.); liyuguang@126.com (Y.L.)

**Abstract:** Ensuring the stable operation of urban wastewater treatment plants (WWTPs) and achieving energy conservation and emission reduction have become serious problems with the improvement of national requirements for WWTP effluent. Based on a wastewater quality analysis, identification of the contaminant removal, and a simulation and optimization of the wastewater treatment process, a practical engineering diagnosis method for the cyclic activated sludge technology process of WWTPs in China and an optimal control scheme are proposed in this study. Results showed that exceeding the standard of effluent nitrogen and phosphorus due to unreasonable process cycle setting and insufficient influent carbon source is dangerous. The total nitrogen removal rate increased by 9.5% and steadily increased to 67% when agitation was added to the first 40 min of the cycle. Additionally, the total phosphorus (TP) was reduced to 0.27 mg/L after replacing the phosphorus removal agent polyferric sulfate with polyaluminum iron. The corresponding increase in the TP removal rate to 97% resulted in a reduction in the treatment cost by 0.008 CNY/t.

**Keywords:** BioWin model; CAST process; nitrogen and phosphorus removal; process diagnosis; wastewater treatment plants

## 1. Introduction

Water pollution has become a serious problem with the continuous development of urbanization and industrialization in China. The total wastewater discharge was 68.5 billion tons in China in 2012 [1], and the annual wastewater discharge of 480.3 billion tons in 2016 increased to 571.4 billion tons in 2020. WWTPs, the largest wastewater treatment unit, are responsible for the treatment of domestic sewage and industrial wastewater. However, unqualified effluent discharged by WWTPs has become a considerable source of water pollution. Therefore, improving the treatment efficiency of WWTPs is crucial to ensure effluent quality and reduce water pollution. In addition, meeting stringent wastewater discharge standards for the effluent of WWTPs has become increasingly difficult with the continuous progress of upgrading, expanding, and constructing new WWTPs in China. Therefore, establishing a feasible operation scheme of WWTPs plays an important role in wastewater treatment.

CAST wastewater treatment technology is an improved process of the sequencing batch reactor (SBR)-activated sludge process and a conventional operation scheme of WWTPs that has been widely used in domestic sewage treatment [2]. A new type of CAST process based on SBR-activated sludge and biological selector was successfully developed in the late 1960s [3]. CAST is a deformation process of SBR that presents unique advantages over other wastewater treatment processes, such as a short operation cycle, high treatment efficiency, strong impact loading resistance, small footprint, and low infrastructure cost [4,5].

However, wastewater treatment processes typically adopt the constant parameter control mode, which relies on empirical data, due to the lack of relevant knowledge on WWTPs of some technicians. At present, research on wastewater treatment process optimization in China is basically at the laboratory scale or in the process of upgrading, and studies on the improvement of the practical CAST-process WWTPs are limited. For WWTPs with different treatment processes, diagnostic methods will be conducive to practical application and suitable for extensive promotion.

The operation problems of CAST-process WWTPs mainly focus on the inability to remove nitrogen and phosphorus pollutants stably and efficiently. Therefore, CAST-process WWTPs located in China are used as research objects in this study. This work primarily investigates and analyzes the difficulty in stable nitrogen and phosphorus pollutant removal by establishing the method of process diagnosis. Finally, the results of the static simulation experiment, optimized operation regulation, and functional microbial community structure analysis will provide technical guidance for the stable operation of CAST-process WWTPs. This study establishes a diagnostic method and carries out the optimized operation which is expected to significantly reduce pollutant emissions and achieve the requirements of energy saving and consumption reduction.

## 2. Materials and Methods

### 2.1. Diagnostic Method

As shown in Figure 1, water quality parameter characteristics of $BOD_5/COD$, $BOD_5/TN$, and $BOD_5/TP$ are analyzed in detail according to the historical influent water quality and 24-h influent water quality statistics of the WWTP with the CAST process. The analysis of the removal effect of organic matter and nitrogen and phosphorus pollutants can infer the operation efficiency of the biochemical reaction tank and microbial activity of activated sludge. The arrangement of sampling points and the reduction effect of characteristic pollutants in WWTPs demonstrated that the stability of organic matter, nitrogen, and phosphorus removal is the key factor affecting the stable operation of WWTPs. The identification of pollutant characteristics can be used to analyze the difficulty in the stable discharge of pollutants initially. In addition, the main water quality parameters affecting the stable operation of WWTPs can be identified by analyzing variations in basic water quality parameters of the effluent. The laboratory-scale simulation of the CAST process was then carried out to establish the optimization method of WWTPs. The CAST process was verified and calibrated on the basis of the BioWin model to establish the optimal operation strategy of WWTPs.

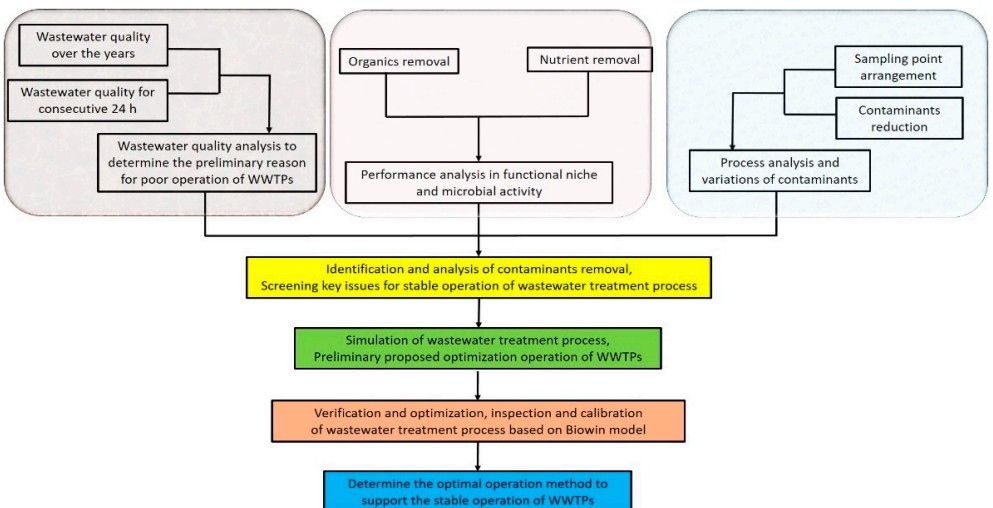

**Figure 1.** Diagnostic method for the operation of WWTPs.

*2.2. Sampling Point and Index Analytical Methods*

2.2.1. Sampling Points in WWTPs

The treatment capacity of the CAST-process WWTP located in China is 25,000 t/d. The process flow of the WWTP is coarse grid → lift pump → fine grid → swirling sand settling → anaerobic hydrolysis → CAST process → inclined plate sedimentation → turntable filtration → UV disinfection. The CAST process runs for 4 h, including 2 h of influent and aeration, 1 h of settling and sedimentation, and 1 h of effluent and sludge discharge. The specific sampling points are shown in Figure 2. According to the CAST process, sampling points are arranged along with the pollutant removal process, including pretreatment, biochemical treatment, and advanced treatment processes. Four samples were collected in each treatment process, and the water quality COD, TN, TP, ammonia nitrogen, nitrate, $PO_4^{3-}-P$, and MLVSS were measured.

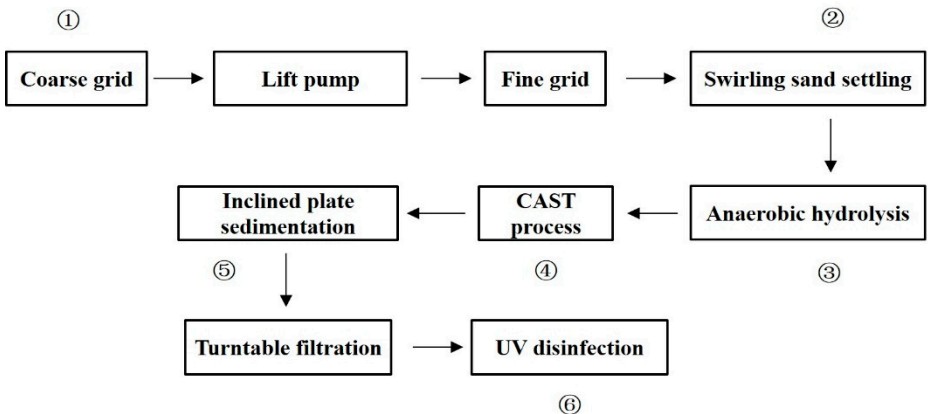

**Figure 2.** Sampling points in the CAST-process WWTP.

2.2.2. Analytical Methods

Chemical oxygen demand (COD), nitrate, total nitrogen (TN), ammonia nitrogen, $PO_4^{3-}-P$, total phosphorus (TP), and mixed-liquor volatile suspended solids (MLVSSs) were measured according to the standard method [6]. The DNA of the activated sludge was extracted by bacterial genomic DNA extraction kits (ABigen, Beijing, China), and the extracted rDNA was then amplified via PCR using the universal primers 341F and 785R. Additionally, the positive clones after evaluation by electrophoresis were sequenced on an Illumina MiSeq platform at Genergy Biotechnology (Shanghai, China).

Nitrification Rate

Activated sludge (4 L) with MLVSS (3500 mg/L) was sampled from the CAST process of WWTPs, and dissolved oxygen (DO) was maintained at 2–3 mg/L. $NH_4Cl$ (1.20 g) and $NaHCO_3$ (1.00 g) (supply alkalinity for nitrification) were added to the nitrification system, and the nitrate concentration was measured every 10 min. The linear regression based on the variation of nitrate concentration was obtained and the nitrification rate was calculated with the biomass retention of activated sludge.

Denitrification Rate

Activated sludge (4 L) with MLVSS (3500 mg/L) was sampled from the CAST process of the WWTP, and a moderate carbon source was added while stirring to reduce the DO below 0.5 mg/L. $KNO_3$ (1.20 g) and NaAc (1.20 g) were directly added to the denitrification system and the nitrate concentration was measured at 1, 3, 5, 7, 10, 15, 20, 30, 45, 60, 90, and 120 min. Linear regression based on the variation of nitrate concentration was obtained, and the denitrification rate was calculated with the biomass retention of activated sludge.

Denitrification Potential

The denitrification potential was measured to evaluate the actual denitrification performance of activated sludge in the CAST process. Activated sludge (8 L) with MLVSS (3500 mg/L) was sampled from the CAST process of WWTP, and the DO concentration was maintained below 0.3 mg/L. The supply of 1.20 g of $KNO_3$ and sufficient NaAc resulted in the high bioactivity of activated sludge. The nitrate concentration was measured at 1, 3, 5, 7, 10, 15, 20, 30, 45, 60, 90, and 120 min. Linear regression based on the variation of nitrate concentration was obtained, and the denitrification potential was calculated with the biomass retention of activated sludge.

Phosphorus Release Rate

Activated sludge (8 L) with MLVSS (3500 mg/L) was sampled from the CAST process of the WWTP and activated sludge was washed with distilled water three times to remove the nitrate interference for denitrification. NaAc (1.50 g) was provided to maintain a sufficient carbon source and reduce the DO contention below 0.1 mg/L. Phosphate concentration was measured every 10 min, and the linear regression based on the variation of the $PO_4^{3-}$–P concentration was obtained. The phosphorus release rate was calculated with the biomass retention of activated sludge.

Static Simulation for Enhancing Nitrogen Removal

a.      The nitrogen removal performance of the CAST process

Sludge (6.4 L, biomass of 3500 mg/L) and influent (1.6 L) were added to the laboratory-scale CAST process on the basis of a sludge/wastewater ratio of 4. The operating mode was as follows: 0–120 min of aeration (0–40 min, DO is at 0.5–1.0 mg/L; 40–120 min, DO is above 2.0 mg/L) and 120–240 min of settling and static discharge. Samples were taken at 1, 3, 4, 5, 7, 10, 15, 25, 40, 60, 75, 90, 120, 180, and 240 min to determine the denitrification and nitrification performance according to the variations of ammonia nitrogen and nitrate concentration.

b.      Static simulation for enhancing nitrogen removal

The operation mode was regulated to enhance the denitrification performance as follows: static influent in the first 40 min with stirring agitation, aeration with different DO concentrations in 40–120 min (40–60 min, DO is at 0.5–1.0 mg/L; 60–120 min, DO is above 2.0 mg/L), and 120–240 min of settling and static discharge. Samples were taken at 1, 3, 4, 5, 7, 10, 15, 25, 40, 60, 75, 90, 120, 180, and 240 min to determine the denitrification and nitrification performance according to variations of ammonia nitrogen and nitrate concentration.

Screening of Phosphorus Removal Chemicals

The influent (2 L) was sampled and then placed in three beakers before adding three types of phosphorus removal chemicals. The coagulation condition program was set to 1 min for fast stirring (500 r/min), 5 min for moderate stirring (100 r/min), 5 min for slow stirring (30 r/min), and 15 min for precipitate. Finally, supernatants were collected for the measurement of TP, $PO_4^{3-}$–P, and COD.

## 3. Results and Discussion

### 3.1. Diagnosis for Nitrogen Removal in the CAST Process

3.1.1. Analysis of Influent and Effluent Nitrogen

Data analysis of Figure 3 demonstrated that the highest value of influent TN is about 49.1 mg/L, the lowest value is below 19.9 mg/L, and the average value of influent TN is approximately 35.0 mg/L, thereby showing significant fluctuations. The cumulative distribution of the TN concentration in the effluent shown in Figure 4 indicated that nearly 20% of the effluent TN value exceeds the discharge standard.

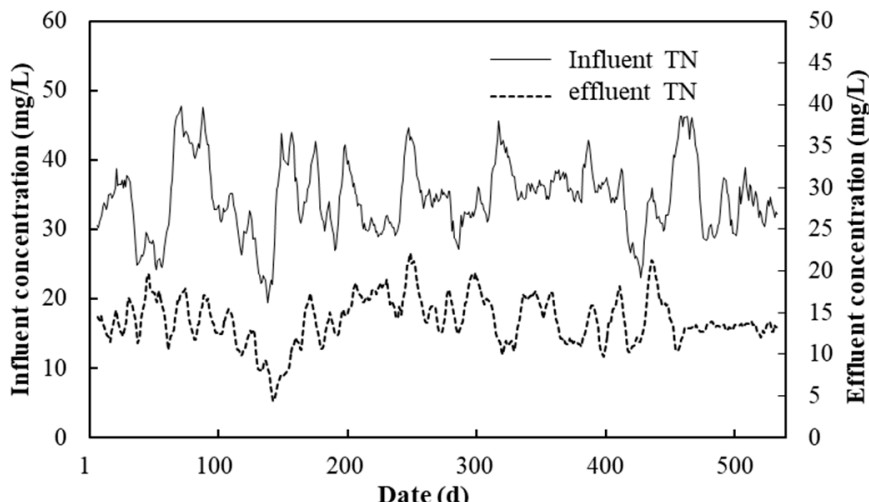

**Figure 3.** Influent and effluent of total nitrogen in the CAST process (discharge standard: 15.0 mg/L).

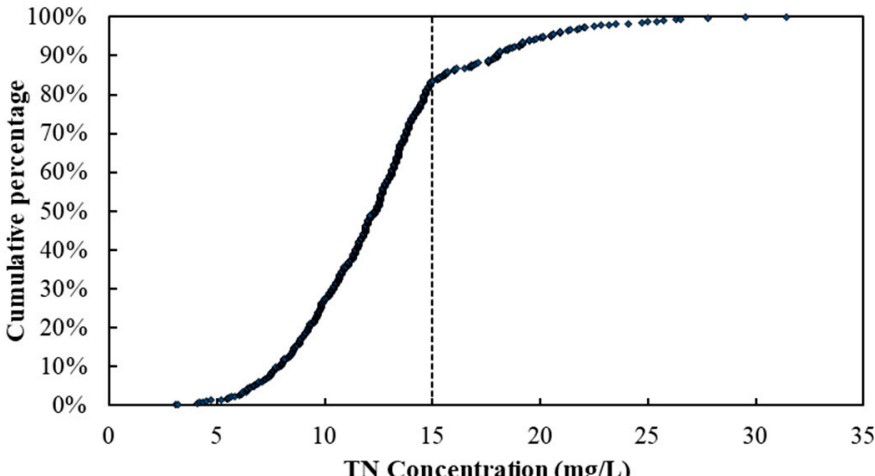

**Figure 4.** Total nitrogen cumulative distribution of the effluent in the CAST process.

As depicted in Figure 5, the highest concentration of influent ammonia nitrogen is about 31.0 mg/L, the lowest value is 12.5 mg/L, and the average value is approximately 20.7 mg/L. The peak value of effluent ammonia nitrogen concentration appears in winter likely due to the low bioactivity of activated sludge [7], but the effluent ammonia nitrogen concentration is still greater than the discharge standard. In addition, the effluent ammonia nitrogen was less than 3.0 mg/L with the increase in temperature.

$BOD_5/TN$ is the main indicator for identifying the capability of biological nitrogen removal in WWTP [8]. Theoretically, $BOD_5/TN > 2.86$ indicates that TN can be effectively removed in any type of wastewater treatment process due to the presence of DO competing with nitrate for an electron donor, and the corresponding $BOD_5/TN$ ratio must be higher than 5.0 to achieve efficient TN removal in the practical operation of WWTPs. Figure 6 shows that the average value of $BOD_5/TN$ is approximately 3.0 and the probability of $BOD_5/TN > 5$ is only 5%. This finding implied that the removal of biological nitrogen is unstable in the CAST process.

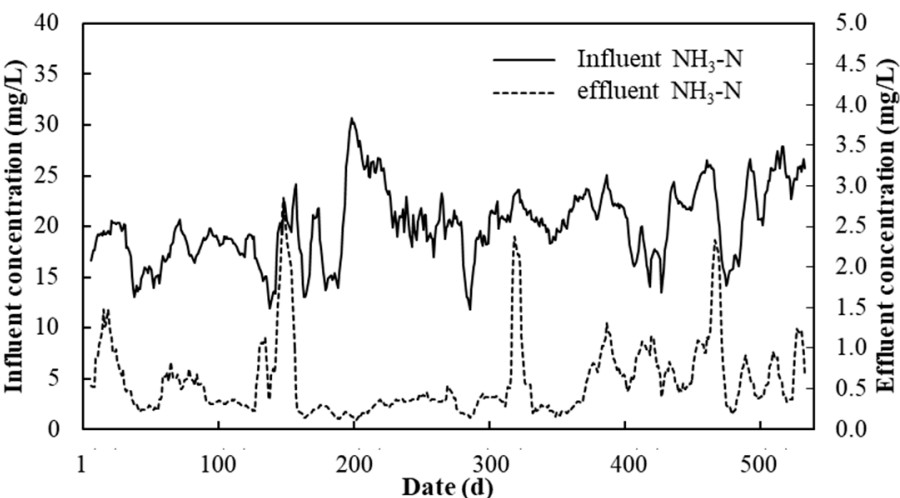

**Figure 5.** Influent and effluent of ammonia nitrogen in the CAST process (discharge standard: 5.0 mg/L).

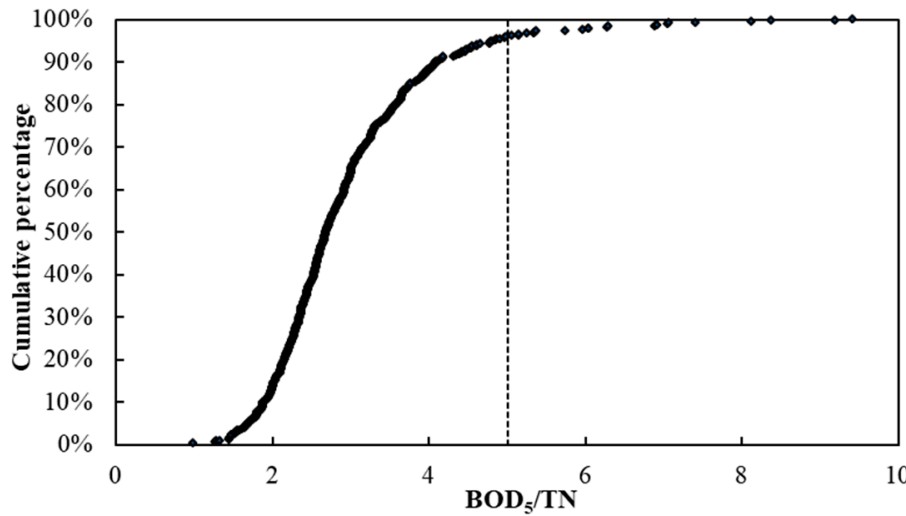

**Figure 6.** $BOD_5/TN$ cumulative distribution in the CAST process.

### 3.1.2. Microbial Activity of Activated Sludge for Nitrogen Removal
### Nitrification Rate

The nitrogen removal pathway mainly includes nitrification and denitrification processes. Ammonia nitrogen is oxidized by ammonia-oxidizing bacteria (AOB), and nitrite can be subsequently oxidized by nitrite-oxidizing bacteria (NOB) into nitrate. Finally, nitrate can be effectively reduced by denitrifiers to $N_2$ when a sufficient carbon source is provided [9,10]. Ammonia nitrogen is oxidized to nitrite and nitrate and the alkalinity generated by denitrification is consumed under aerobic conditions. Therefore, the pH value of the entire activated sludge system is stably maintained. The high sensitivity of nitrifying bacteria to the pH value can provide the basis for the smooth progress of the whole denitrification process [11]. Figure 7a,b show that the nitrification rates of activated sludge are 4.36 and 5.24 $mgNO_3^- -N/(gVSS \cdot h)$, with an MLVSS of 3500 and 3000 mg/L, respectively. The results indicated that the low biomass of activated sludge in the CAST process can tolerate high ammonia nitrogen loading. This finding is consistent with the low effluent ammonia nitrogen during the WWTP operation.

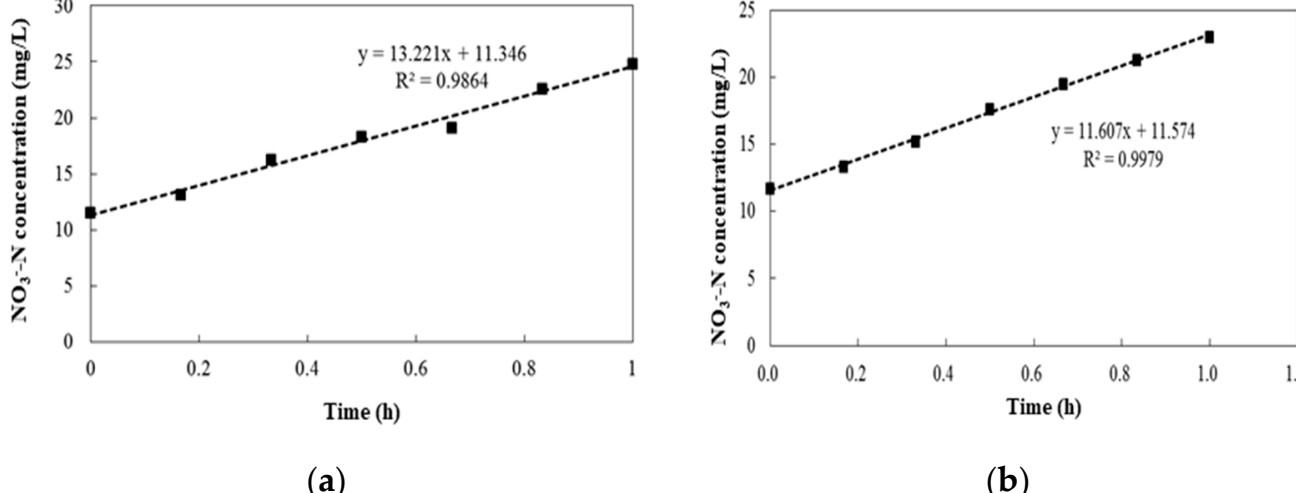

**Figure 7.** The nitrification rate of activated sludge in the CAST process with different biomass amounts. (**a**): MLVSS of 3500 mg/L; (**b**): MLVSS of 3000 mg/L; average nitrification rate is below 4.0 mg/gVSS·h in China.

Denitrification Rate

The denitrification reaction is a process in which denitrifying bacteria reduce nitrite and nitrate to gaseous nitrogen ($N_2$) under anoxic conditions [7]. Denitrification rate, denitrification potential, and the endogenous denitrification rate were measured to investigate the denitrification performance of activated sludge in the CAST process. As shown in Table 1, the denitrification rate of activated sludge in the CAST process can be divided into three periods: a carbon-source rapid utilization period, a carbon-source slow utilization period, and endogenous denitrification [12]. The denitrification rate (Figure 8a) and denitrification potential (Figure 8b) are low, and only 5.6 mg/L of nitrate can be removed within 1 h by providing sufficient carbon sources. The results of influent $BOD_5/TN$ analysis (Figure 6) demonstrated that rapid carbon sources within the influent are few and the endogenous denitrification performance (Figure 8c) can be maintained. The low denitrification rate is caused by the following: (1) minimal high-quality carbon sources existing in the influent are not conducive to the denitrifying bacteria and maintaining high microbial activity [13] and (2) the absence of an anoxic period in the CAST operation cycle results in the poor bioactivity of denitrification bacteria.

**Table 1.** The denitrification rate of activated sludge in the CAST process.

| Group | First-Stage Denitrification Rate ($mgNO_3{}^-$–N/(gVSS·h)) | Second-Stage Denitrification Rate ($mgNO_3{}^-$–N/(gVSS·h)) | Third-Stage Denitrification Rate ($mgNO_3{}^-$–N/(gVSS·h)) |
|---|---|---|---|
| Denitrification rate | 1.16 | 0.55 | 0.25 |
| Denitrification potential | 2.84 | / | / |
| Endogenous denitrification | 0.39 | / | / |

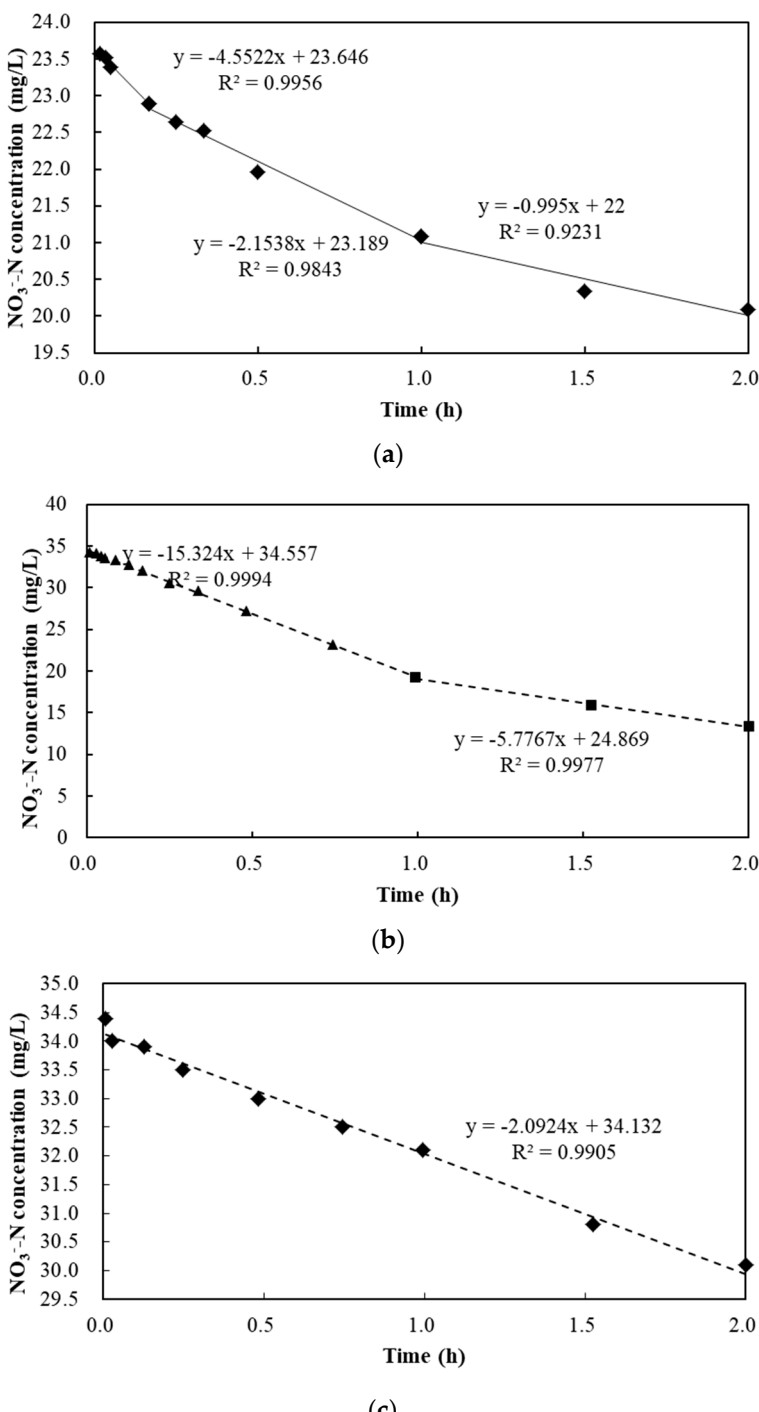

**Figure 8.** (**a**) The denitrification rate of activated sludge in the CAST process (average denitrification rate is below 3.0 mg/gVSS·h in China); (**b**) The denitrification potential rate of activated sludge in the CAST process; and (**c**) The endogenous denitrification rate of activated sludge in the CAST process.

### 3.1.3. Simulation and Optimization for TN Removal

The nitrification rate of activated sludge is high, but the denitrification performance is remarkably low under the current process operating conditions. Figure 9a shows the nitrate variation by setting the agitation process in the first 40 min (enhanced denitrification mode) before aeration. The experimental results showed that the removal of 3.0 mg/L of nitrate achieves satisfactory denitrification performance. However, 4.8 mg/L of nitrate was generated when aeration was applied in the first 40 min (Figure 9a). The initial concentration of ammonia nitrogen in the simulated field mode is 8.4 and 8.8 mg/L, which

dropped to 0.8 mg/L after 60 min and 0.6 mg/L after 90 min, respectively (Figure 9b). The results showed that the effluent ammonia nitrogen can be less than 1.0 mg/L, although the aeration time is shortened. Regulating the original aeration period to 40 min of agitation and 80 min of aeration in the CAST process is conducive to enhancing nitrate reduction and TN removal.

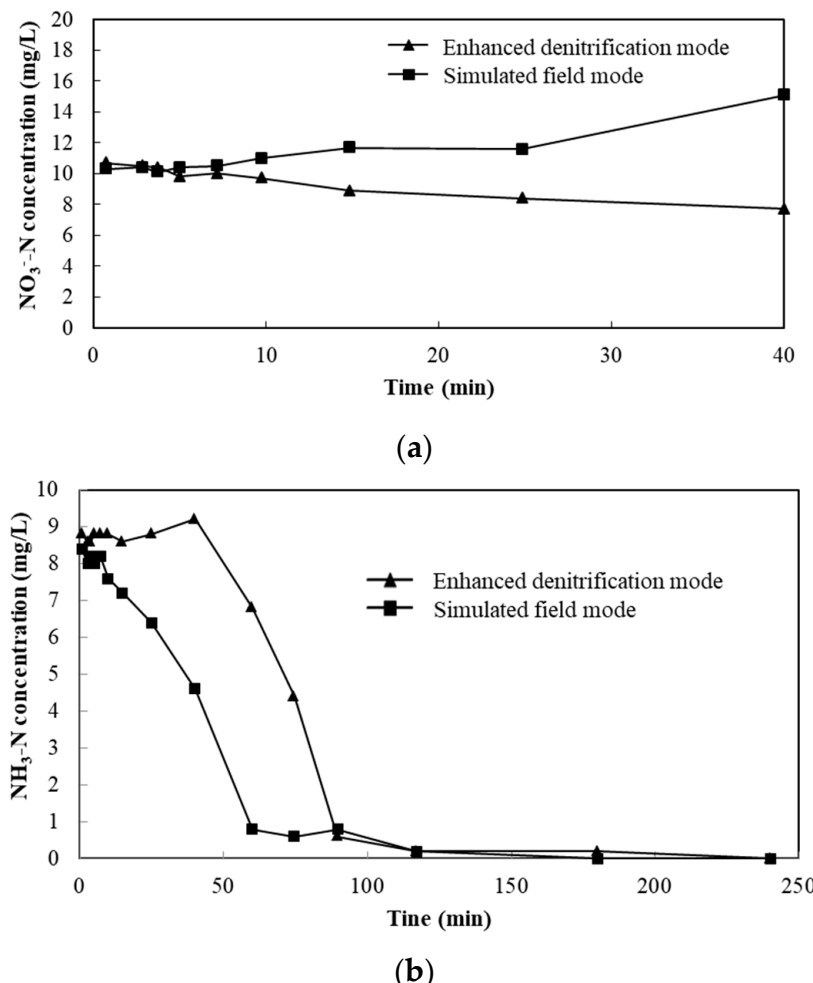

**Figure 9.** (**a**) Nitrate variation by simulation and optimization in the CAST process; (**b**) Ammonia nitrogen variation by simulation and optimization in the CAST process.

3.1.4. Modeling and Optimization Control

Process Operation Parameters

A CAST-generalized model based on BioWin is established according to the practical operation conditions of the WWTP (Figure 10). Pretreatment equipment coarse grid, fine grid, and swirling sand settling are omitted from the model, and the post-treatment equipment is replaced with a sedimentation tank. The effective volume of the selection zone and biochemical reaction tank is 1012 and 7260 $m^3$, respectively, with a water depth of 6.0 m. In addition, the specific influent of COD, total Kjeldahl nitrogen (TKN), TP, nitrate, and inorganic solid suspension (ISS) are 294, 32.5, 7.5, 2.0, and 100 mg/L, respectively, with a pH value and temperature of 7.3 and 20 °C, respectively.

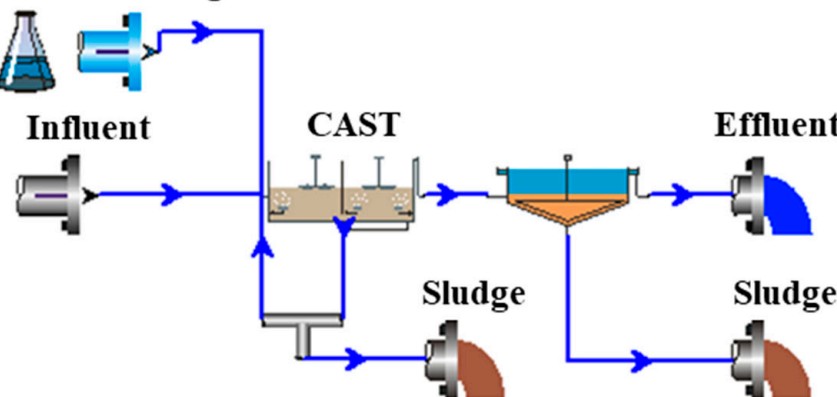

**Figure 10.** Generalized model of the CAST process based on BioWin.

Optimization of Reflux Ratio

The unique reaction zone of the CAST process can be used for biological selection to improve the stable operation. The mixed liquid in the main reaction zone is refluxed to the selection zone in the influent stage to facilitate the denitrification property in the biological selection zone. Therefore, the optimal reflux ratio can be obtained by comparing the denitrification rates. At present, the reflux ratio of the CAST process is 60%, and the sludge discharge is 294 m$^3$/d. According to the simulation results (Table 2), effluent COD, BOD$_5$, and TN concentrations decrease as the reflux ratio of the mixed solution increases. Meanwhile, the effluent ammonia nitrogen will increase to a certain extent. Therefore, the reflux ratio can be increased appropriately under acceptable nitrification conditions.

**Table 2.** The simulation of different reflux ratios in the CAST process.

| Reflux Ratio | Outlet Concentration (mg/L) | | | | |
| --- | --- | --- | --- | --- | --- |
| | COD | BOD$_5$ | Ammonia Nitrogen | TN | TP |
| 10% | 26.90 | 2.22 | 1.24 | 10.62 | 0.01 |
| 20% | 26.54 | 1.93 | 1.41 | 10.22 | 0.01 |
| 60% | 25.97 | 1.48 | 2.07 | 10.07 | 0.01 |
| 100% | 25.80 | 1.24 | 2.31 | 9.82 | 0.01 |

Simulation of Agitation Period

According to the simulation and optimization results, adding a period of agitation before aeration can improve the nitrogen removal in the CAST process. The simulation results are listed in Table 3. Increasing the agitation time to 40 min can effectively reduce the effluent TN by 1.97 mg/L, while the effluent COD, ammonia nitrogen, and TP can still reach the discharge standard stably.

**Table 3.** The simulation results from different operation periods of the CAST process.

| CAST Operation (min) | | | | Effluent Concentration (mg/L) | | | | |
| --- | --- | --- | --- | --- | --- | --- | --- | --- |
| Agitation | Aeration | Precipitation | Discharge | COD | BOD$_5$ | Ammonia Nitrogen | TN | TP |
| 0 | 120 | 60 | 60 | 25.97 | 1.48 | 1.07 | 10.07 | 0.01 |
| 40 | 80 | 60 | 60 | 27.46 | 2.45 | 2.60 | 8.10 | 0.01 |

3.1.5. Effect of Process Optimization and Control

According to the experimental results of simulation optimization, activated sludge modeling, and optimization control analysis, the WWTP changed the CAST process operation cycle by setting 40 min of agitation before aeration, thereby shortening the aeration

time to 80 min. As shown in the TN removal variation in Figure 11, the average TN removal is 57.5% and the current operation mode fluctuates significantly. However, the TN removal increased by a factor of 9.5% and showed stable TN performance when the optimization operation strategy was adopted.

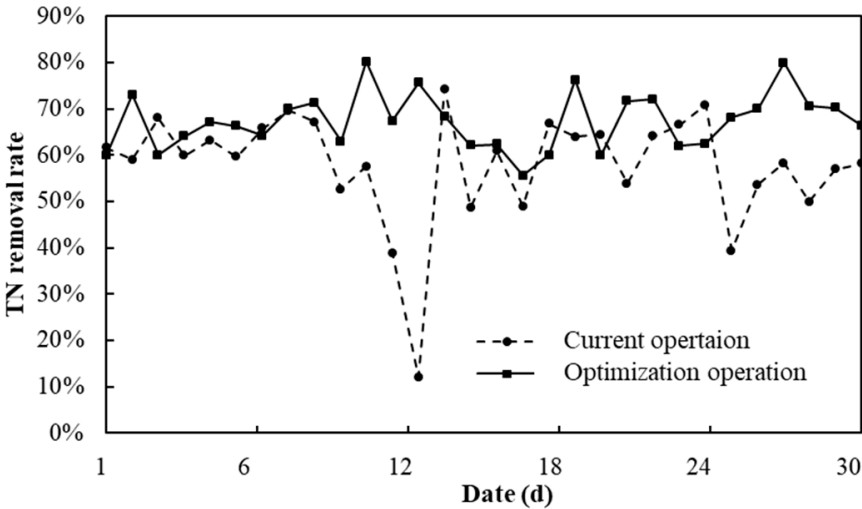

**Figure 11.** TN removal in the CAST process via optimization.

*3.2. Diagnosis for Phosphorus Removal in the CAST Process*

3.2.1. Analysis of Influent and Effluent Phosphorus

Figure 12 shows that the influent TP concentration fluctuates significantly. The highest value is approximately 14.7 mg/L, the lowest value is about 4.1 mg/L, and the average value is 8.9 mg/L. The effluent TP is large, and almost 15% of the effluent TP value exceeds the discharge standard.

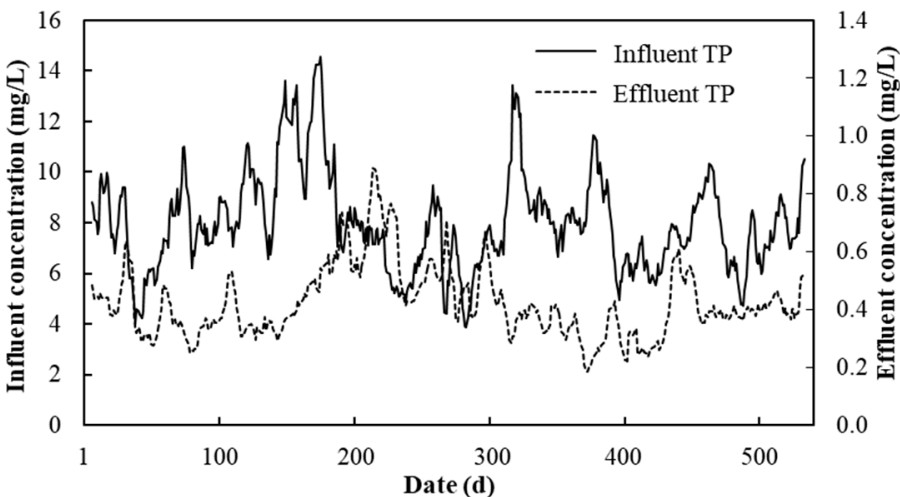

**Figure 12.** The influent and effluent of total phosphorus in the CAST process (discharge standard: 0.5 mg/L).

Influent $BOD_5/TP$ is the main indicator for identifying the biological phosphorus removal capacity. Theoretically, $BOD_5/TP$ should be greater than 17 to achieve complete biological phosphorus removal [14]. According to Figure 13, 80% of the influent $BOD_5/TP$ value of WWTP is less than 17 and the influent C/P is seriously low. The anaerobic stage is the main stage for phosphorus release. Phosphate-accumulating organisms (PAOs) fail to release phosphorus properly without the addition of a carbon source under anaerobic

conditions. In addition, poor phosphorus release can further lead to the unsatisfactory phosphorus uptake performance of PAOs in the aerobic section, and subsequently, decrease the biological phosphorus removal property [15,16].

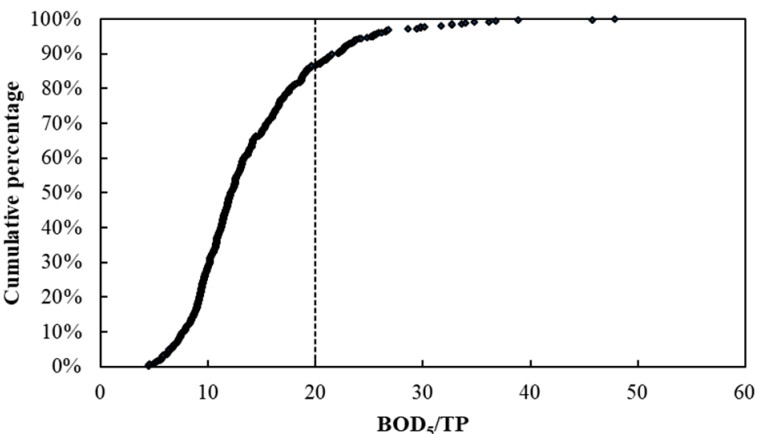

**Figure 13.** $BOD_5/TP$ cumulative distribution in the CAST process.

### 3.2.2. Microbial Activity of Activated Sludge for Phosphorus Removal
#### Phosphorus Release Rate

Biological phosphorus removal is an efficient phosphorus removal technology, mainly through the phosphorus release of PAOs under anaerobic conditions and excessive phosphorus uptake under aerobic conditions; notably, influent phosphorus can be effectively removed through sludge discharge [17]. Anaerobic phosphorus release is a process in which PAOs utilize influent COD under anaerobic conditions and hydrolyze polyphosphorus and intracellular polysaccharides to produce energy and form poly-β-hydroxybutyric acid (PHB). An effective anaerobic phosphorus release reaction can enhance excessive phosphorus uptake under aerobic conditions and achieve efficient phosphorus removal performance by discharging the sludge [18]. Therefore, the anaerobic phosphorus release rate can be employed to characterize the bioactivity of PAOs in activated sludge.

Figure 14 depicts the phosphorus release rate of activated sludge in the CAST process. The phosphorus release rate is only 0.03 $mgPO_4^{3-}P/(gVSS \cdot h)$ with a sludge MLVSS of 3500 mg/L. This significantly low finding showed that the microbial activity of PAOs is highly inhibited, likely due to the absence of an independent anaerobic environment in the CAST process cycle [19] and the inhibition of PAO bioactivity from the addition of chemical phosphorus removal agents [20].

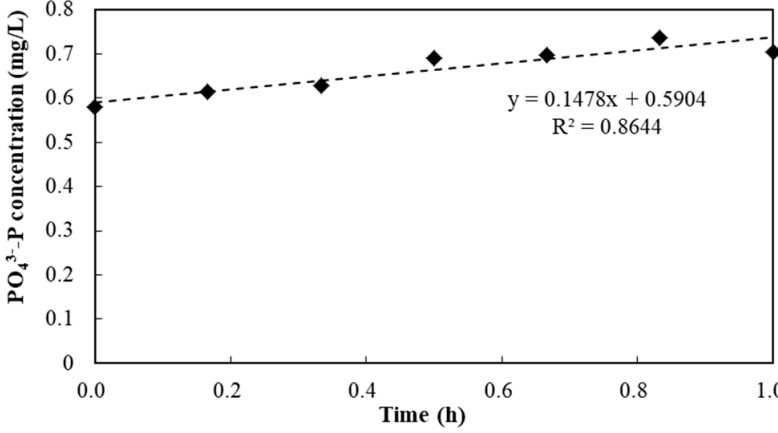

**Figure 14.** The phosphorus release rate of activated sludge in the CAST process.

Microbial Community Structure

Three WWTPs in China were compared to verify the inhibition of phosphorus removal chemicals in activated sludge. The abundance of the microbial community is shown in Figure 15a,b using Global Alignment for Sequence Taxonomy (GAST) processing. Polyferric sulfate was used in HS WWTP, polyaluminum chloride was adopted in MC WWTP, and HD WWTP was set as the control because no chemical phosphorus removal agent was employed.

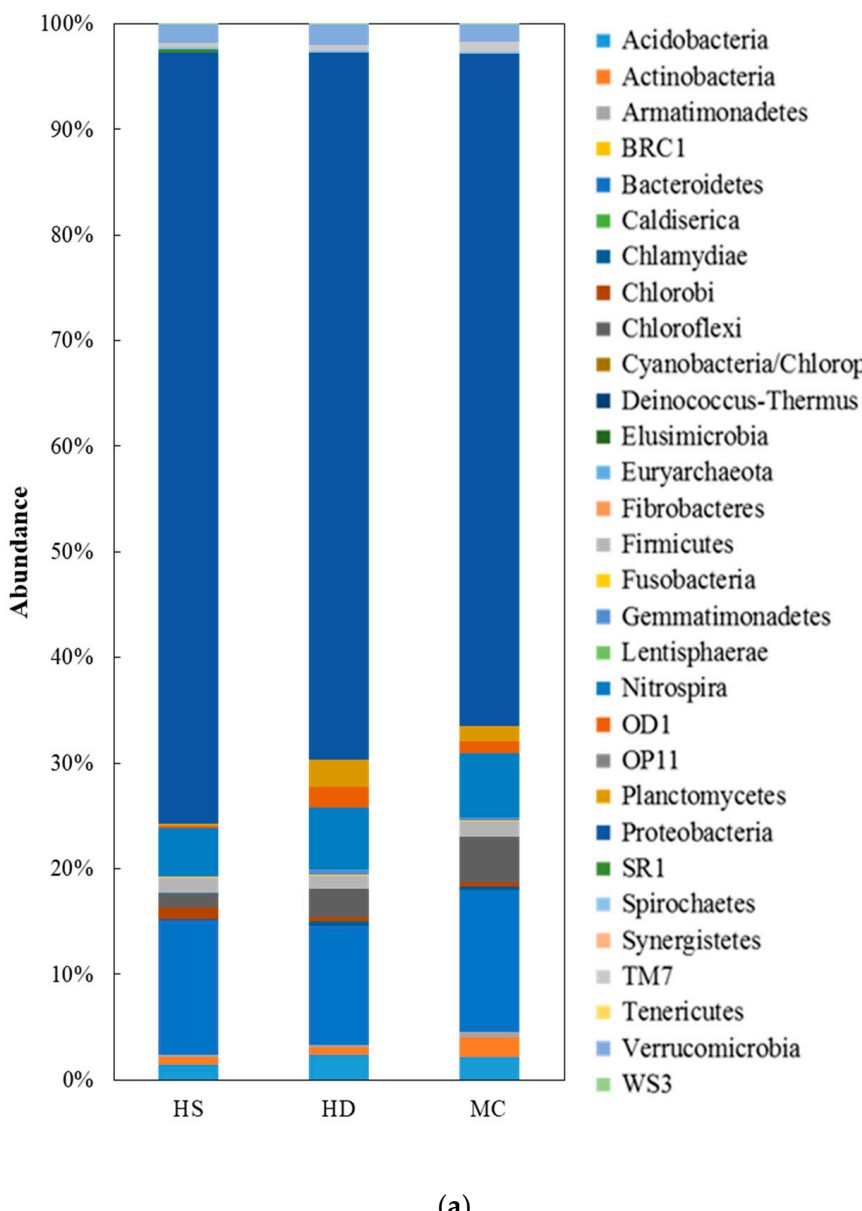

(**a**)

**Figure 15.** *Cont.*

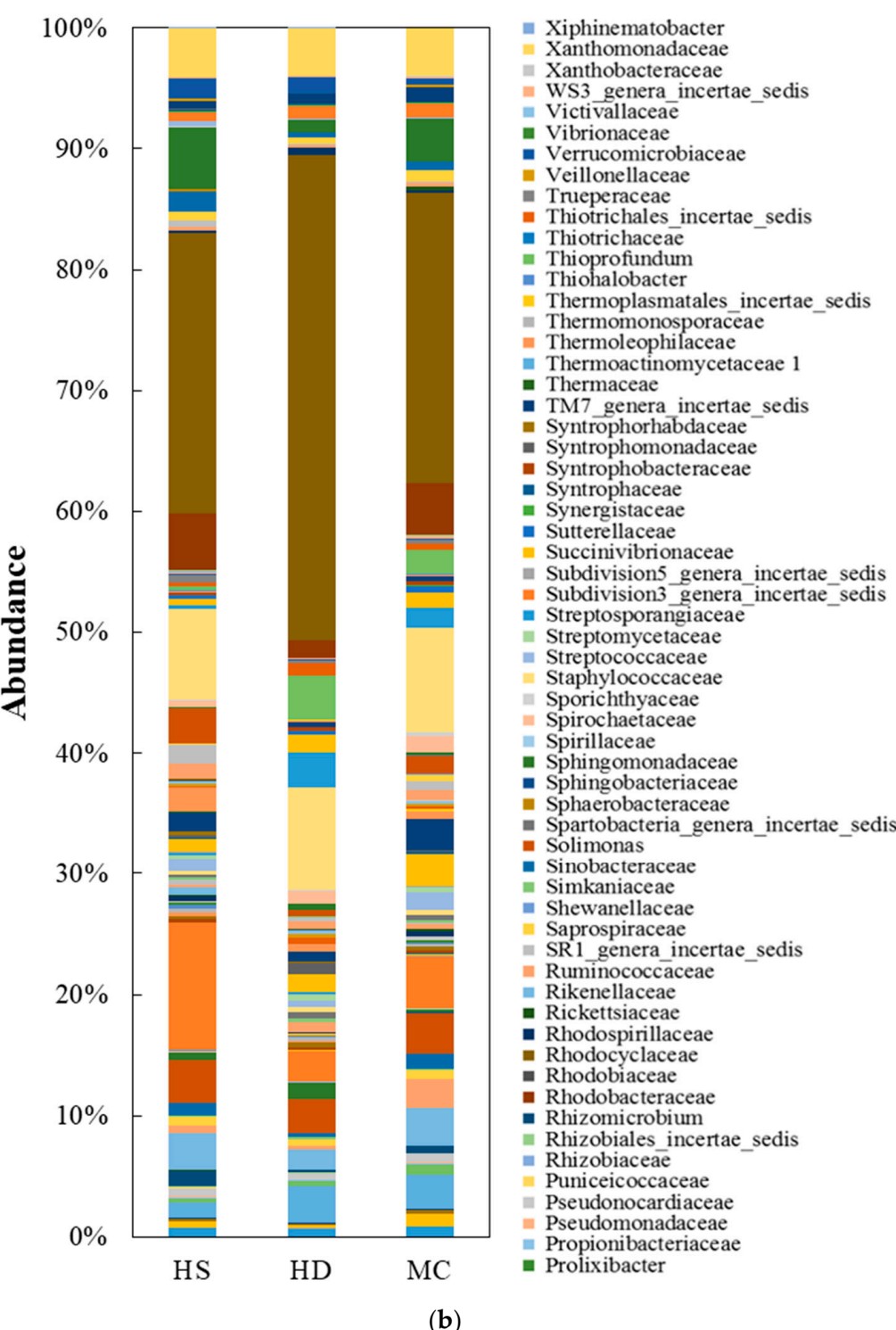

**Figure 15.** (**a**) Microbial community structure at the phylum level; (**b**) Microbial community structure at the family level.

*Proteobacteria* is dominant in three WWTPs with a relative abundance of 63.66%–73.02% [21,22]; other dominant phyla are *Bacteroidetes* (11.22%–13.45%), *Nitrospira* (4.47%–6.04%), and *Acidobacteria* (1.46%–2.39%). The abundance of dominant bacteria classified at the phylum level in three sludge samples is similar, but the abundance of *Planctomycetes* in HD is relatively higher than in the others (Figure 15a). Figure 15b presents the distribution of microbial abundance classified at the family level. The microbial community in three WWTPs presented discrepancies; the relative abundance of the dominant *Rhodocyclaceae*

is 23.21%–40.20%, and the abundance of *Rhodocyclaceae* in HD is significantly high. Approximately 40%–69% of PAOs belong to *Rhodocyclaceae* [23,24]; therefore, PAOs in HD WWTP are abundant, and the bioactivity in HD WWTP is not inhibited by the addition of phosphorus removal chemicals. The microbial flora of activated sludge in three WWTPs was preliminarily analyzed using different chemical phosphorus removal agents which exert a certain impact on the activated sludge flora and mainly influence the abundance of *β-proteobacteria*, *α-proteobacteria*, and *Rhodocyclaceae*. The use of an accurate chemical dosing device and ensuring the precise dosage of chemical agents should be investigated further to reduce energy consumption given that chemical phosphorus removal agents present a certain adverse effect on the activated sludge in WWTPs.

### 3.2.3. Optimization for Phosphorus Removal

Biological phosphorus removal in the CAST-process WWTP fails to meet the discharge requirements for TP removal. However, adding chemical phosphorus removal agents is necessary to assist in biological phosphorus removal. Chemical phosphorus removal refers to the addition of chemicals containing inorganic metals to react with $PO_4^{3-}$–P to form insoluble materials and precipitation [25]. High-valent metal ion agents will form insoluble compounds with $PO_4^{3-}$–P in wastewater, and $Fe^{3+}$, $Fe^{2+}$, and $Al^{3+}$ are widely adopted [26]. Polyaluminum iron, polyferric sulfate, and polyaluminum chloride have been commonly used as chemical phosphorus removal agents in WWTPs.

### Chemical and Biological Phosphorus Removal

Different types of chemical phosphorus removal agents were added in accordance with the practical operating conditions. Moreover, polyacrylamide (PAM) coagulation is generally used in the sludge dewatering process; therefore, the residual PAM will remain in wastewater. PAM is a commonly used type of nonionic polymer flocculant with a molecular weight of 1.5 to 20 million. PAM molecules can bridge the suspended particles dispersed in the solution to form larger flocs among the particles and present a strong flocculation effect.

The analysis of Figure 16a,b showed that concentrations of TP and $PO_4^{3-}$–P in the supernatant significantly decrease with the increase in the phosphorus removal agent dosage. The polyaluminum iron shows the best removal effect on TP and $PO_4^{3-}$–P, followed by polyferric sulfate and polyaluminum chloride. TP can be reduced to less than 0.5 mg/L when the dosage of polyaluminum iron reaches 20 mg/L, while polyferric sulfate and polyaluminum chloride must reach 30 mg/L to reduce TP.

### Optimization for Phosphorus Removal Agents

Two additional types of chemical phosphorus removal agents in established WWTPs are biological reactions and influent tanks (before entering the biological reaction tank). Adding chemical phosphorus removal agents to the influent tank may be considered given that chemical phosphorus removal agents will exert a certain impact on biological phosphorus removal.

As depicted in Figure 17a–c, the concentration of TP, $PO_4^{3-}$–P, and COD in the supernatant significantly decreases with the increase in the phosphorus removal agent dosage. Polyaluminum iron presents the best removal effect on TP and $PO_4^{3-}$–P, followed by polyferric sulfate and polyaluminum chloride. In addition, polyaluminum chloride shows high COD removal capacity, followed by polyaluminum iron, while polyferric sulfate demonstrates poor COD removal performance.

TP can be less than 0.5 mg/L when the dosage of polyaluminum iron is 20 mg/L. However, the effluent TP cannot be reduced to less than 0.5 mg/L when the concentration of polyferric sulfate and polyaluminum reaches a maximum of 30 mg/L. The addition of polyaluminum chloride and polyferric sulfate to the influent presents poor COD removal capability, and the removal of TP and $PO_4^{3-}$–P is still unsatisfactory. Considerable COD removal after the addition of polyaluminum iron is not conducive for subsequent biological treatment, but the removal effect of TP and $PO_4^{3-}$–P is improved.

The addition of phosphorus removal agents to the influent can achieve satisfactory phosphorus removal; however, its simultaneous removal of particle COD is not conducive to subsequent biological treatment. The results showed that polyaluminum iron shows the optimal phosphorus removal efficiency with an effluent TP below 0.5 mg/L at a dosage of 20 mg/L and the addition of PAM can significantly increase the removal rate of TP, $PO_4^{3-}-P$, and COD. The consumption of polyaluminum iron can decrease by a factor of 25% after adding 0.05 mg/L of PAM. PAM generally presents strong bridging and adsorption performance [27], integrates the advantages of aluminum and iron coagulants, and its synergistic performance demonstrates a fast reaction speed, large floc formation, fast molding, satisfactory activity, and filtration advantages.

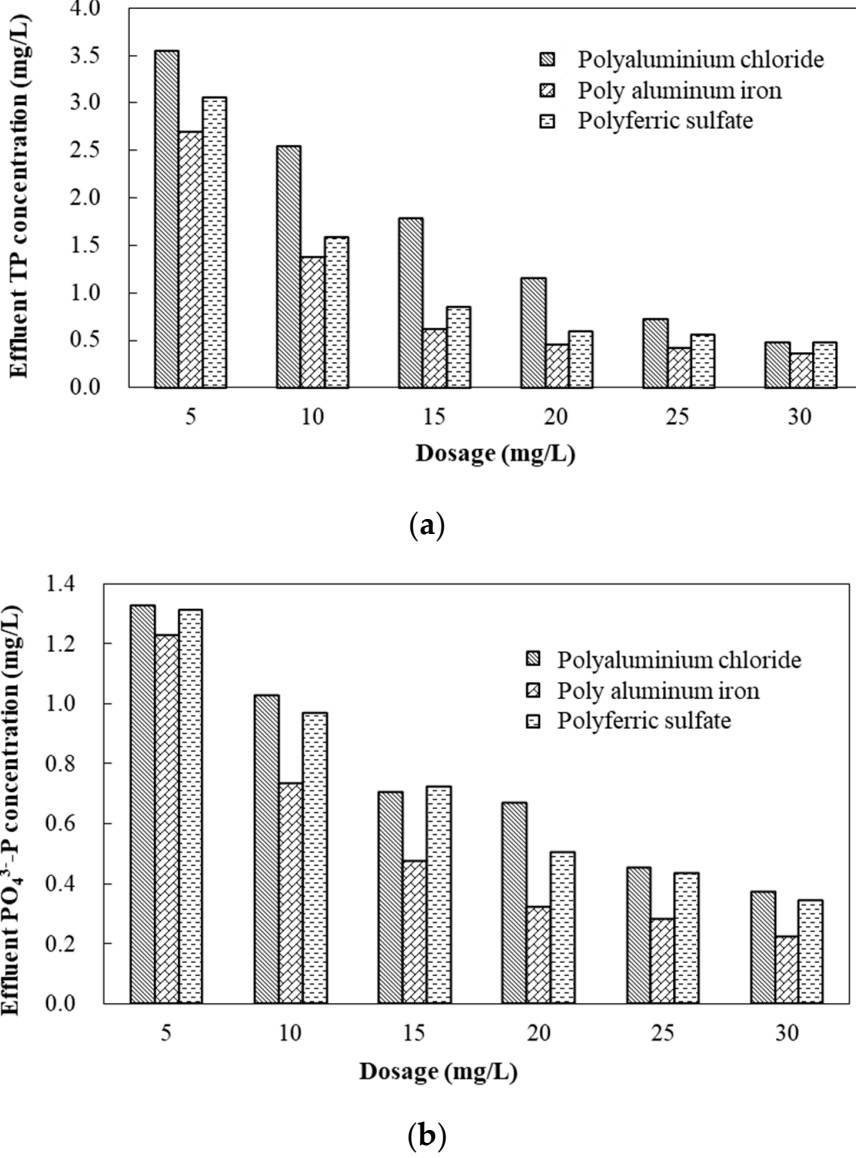

(a)

(b)

**Figure 16.** (**a**) The effect of phosphorus removal agents on TP removal; (**b**) The effect of phosphorus removal agents on $PO_4^{3-}-P$ removal.

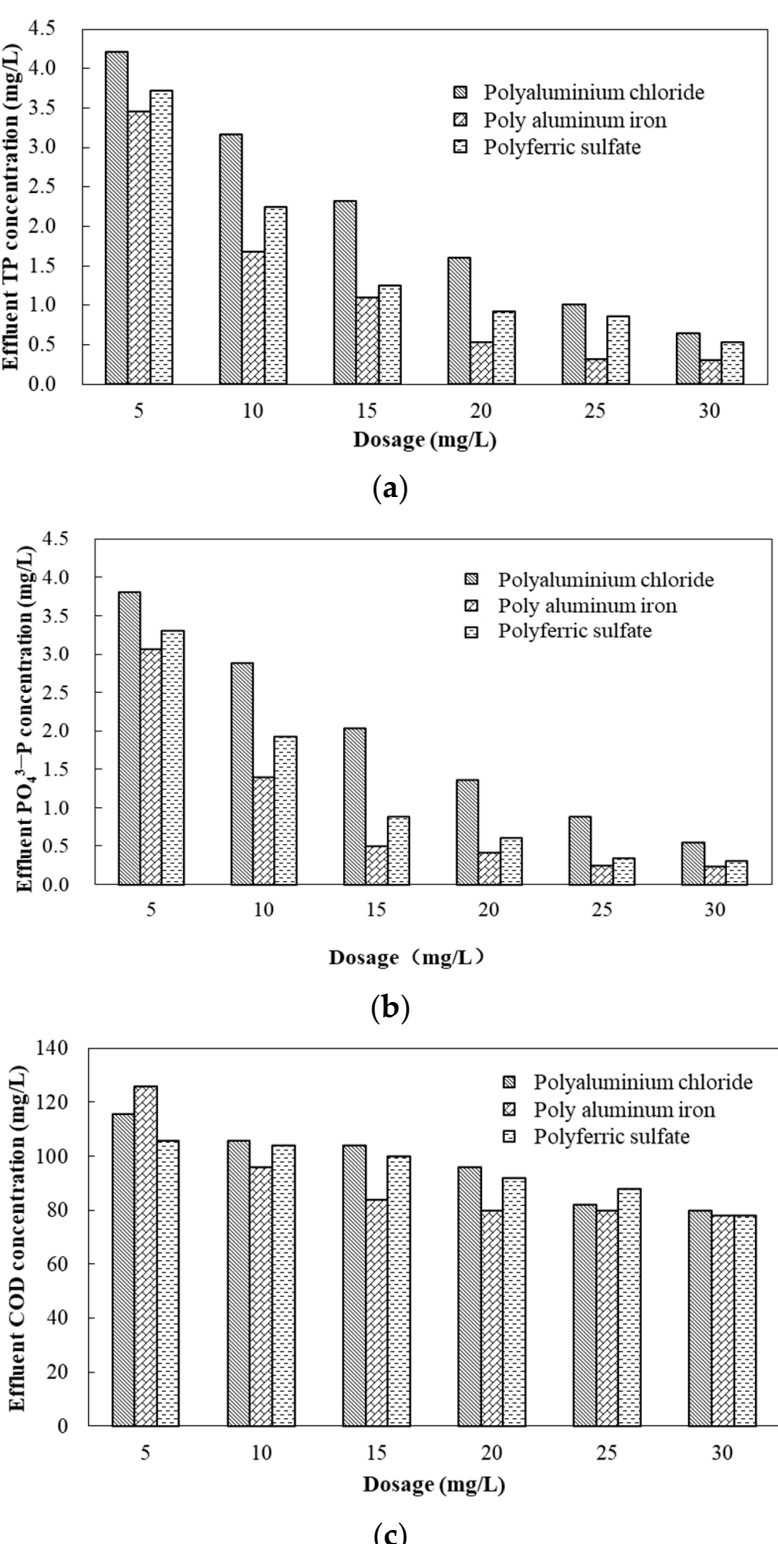

**Figure 17.** (**a**) The effect of phosphorus removal agents on TP in the influent tank; (**b**) The effect of phosphorus removal agents on $PO_4^{3-}$–P in the influent tank; and (**c**) The effect of phosphorus removal agents on COD in the influent tank.

3.2.4. Effect of Process Optimization and Control

According to the experimental results of simulation optimization analysis, the WWTP changed the CAST process by replacing phosphorus removal agents from polyferric sulfate with polyaluminum iron. As shown in Figure 18, TP removal of the CAST process reached

97%, which is about 4% higher and more stable than that in the current operation, after changing the agent. Furthermore, the specific TP removal increased from 0.04 mgTP/mg to 0.05 mgTP/mg, and the reduced consumption of polyaluminum iron from 5 t/d to 4 t/d resulted in the reduction in the treatment cost by 0.008 CNY/t.

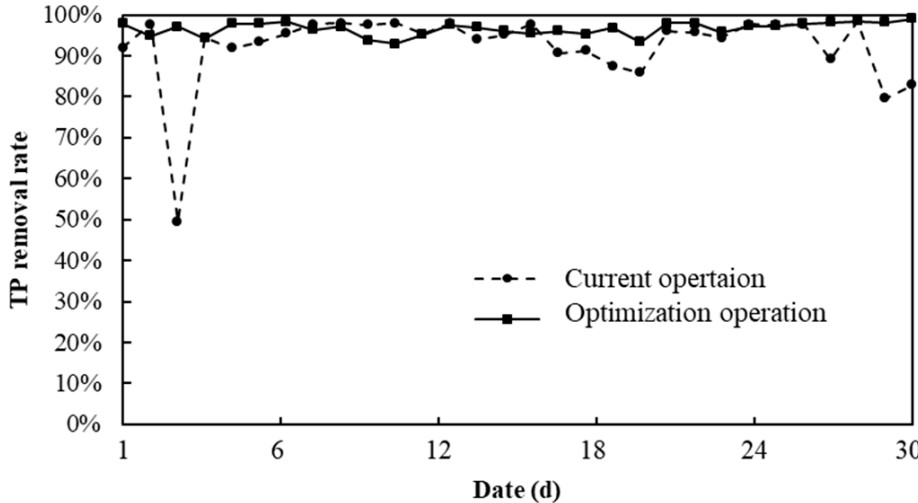

**Figure 18.** TP removal in the CAST process via optimization.

## 4. Conclusions

Low influent $BOD_5/TN$ and $BOD_5/TP$ lead to poor TN and TP removal performance, and additional carbon source and chemical phosphorus removal agents were provided to enhance TN and TP removal in the CAST-process WWTP. A diagnostic method for the optimal operation of WWTPs was established by setting 40 min of agitation before aeration. The average TN removal increased to 67.0% and showed stable TN performance after WWTP optimization. The microbial activity of *Rhodocyclaceae* was inhibited by the addition of chemical phosphorus removal agents. Therefore, the type of chemical phosphorus removal agents must be preliminarily investigated on the basis of the dosage and TP and COD removal properties. TP removal of the CAST process reached 97% after changing the agent, and the increase in the specific TP removal to 0.05 mgTP/m resulted in a reduction in the treatment cost by 0.008 CNY/t.

**Author Contributions:** Conceptualization, Y.L.; methodology, C.L. and K.Q.; investigation, C.L. and K.Q.; writing—original draft preparation, C.L. and K.Q.; writing—review and editing, Y.L.; supervision, Y.L.; project administration, C.L. and Y.L.; funding acquisition, C.L. All authors have read and agreed to the published version of the manuscript.

**Funding:** Funding was provided by the Improvement of scientific research projects for central scientific institutions (118011000000210005) and the Wuxi Innovation and Entrepreneurship Program for Science and Technology (M20211003).

**Institutional Review Board Statement:** Not applicable.

**Informed Consent Statement:** Not applicable.

**Acknowledgments:** The authors thank Li Ji, Wang Yan, and Ruan Zhiyu at Jiangnan University for their supporting data collection and comments on the study. The authors gratefully acknowledge the financial support provided by the Improvement of scientific research projects for central scientific institutions (118011000000210005) and the Wuxi Innovation and Entrepreneurship Program for Science and Technology (M20211003).

**Conflicts of Interest:** The authors declare no conflict of interest.

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
