# Peer review of "Diagnostic Method for Enhancing Nitrogen and Phosphorus Removal in Cyclic Activated Sludge Technology (CAST) Process Wastewater Treatment Plant"

_water, doi:10.3390/w14142253_

Round 1
Reviewer 1 Report
1. The abstract must be revised with details regarding the methodology used, results in numerical form and concluding remarks.
2. A number of abbreviations have been defined in abstract section but not subsequently used in the abstract section. Please remove them.
3. There are numerous grammatical and formatting errors needing extensive revisions.
4. The keyword list needs revisions, it must be arranged alphabetically with no words that are part of manuscript title.
5. Introduction needs to be improved with linkages of the material presented and must contain a clear novelty statement.
6. It is not a proposal “process was operated and the optimal operation strategy of 73 WWTP will be finally established” being a research paper the authors should use grammar properly.
7. The figures resolution must be enhanced
8. Discussion section needs some improvement with proper citations.
9. Revise the conclusion section, in the present form it looks like a carbon copy of the abstract.
10. The references list is not uniform
Author Response
Dear Reviewer #1:
Thank you for your letter and the comments concerning our manuscript entitled "A Diagnostic Method for Enhancing Nitrogen and Phosphorus Removal in CAST Process Wastewater Treatment Plant". (Manuscript No. Water-1753391).
The comments are all valuable and very helpful for revising and improving our paper, as well as the important guiding significance to our researches. We have studied comments carefully and have made corrections which we hope meet with approval. Revised portion are also marking with yellow in the revised paper. The main corrections in the paper and the responds to the comments of reviewer #1 are as following:
Responds to the reviewer’s comments:
- The abstract must be revised with details regarding the methodology used, results in numerical form and concluding remarks.
Response: The abstract had been revised as:
Ensuring the stable operation of urban wastewater treatment plants (WWTPs) and achieving energy conservation and emission reduction have become serious problems with the improvement of national requirements for WWTP effluent. Based on wastewater quality analysis, identification of contaminants removal, simulation and optimization of wastewater treatment process, a practical engineering diagnosis method for the cyclic activated sludge technology process of WWTP in China and an optimal control scheme are proposed in this study. Results showed that exceeding the standard of effluent nitrogen and phosphorus due to unreasonable process cycle setting and insufficient influent carbon source is dangerous. Total nitrogen removal rate increased by 9.5% and steadily increased to 67% when the cycle was agitation was added in the first 40 minutes. The effluent total phosphorus (TP) reduced to 0.27 mg/L after replacing the phosphorus removal agent polyferric sulfate with polyaluminum iron, and the corresponding increase of the TP removal rate to 97% resulted in the reduction of treatment cost by 0.008 CNY/t.
- A number of abbreviations have been defined in abstract section but not subsequently used in the abstract section. Please remove them.
Response: The abstract had been carefully revised.
- There are numerous grammatical and formatting errors needing extensive revisions.
Response: The grammatical and formatting errors had been carefully revised, and the “CERTIFICATE OF ENGLISH EDITING” had been uploaded to the submission system.
- The keyword list needs revisions, it must be arranged alphabetically with no words that are part of manuscript title.
Response: The keywords had been arranged alphabetically.
- Introduction needs to be improved with linkages of the material presented and must contain a clear novelty statement.
Response: The novelty statement of the study had been added in Introduction section.
- It is not a proposal “process was operated and the optimal operation strategy of 73 WWTP will be finally established” being a research paper the authors should use grammar properly.
Response: The sentence had been deleted.
- The figures resolution must be enhanced
Response: The figures resolution had been enhanced accordingly.
- Discussion section needs some improvement with proper citations.
Response: Proper citations had been added in Discussion section.
- Revise the conclusion section, in the present form it looks like a carbon copy of the abstract.
Response: The conclusion had been revised as:
Low influent BOD5/TN and BOD5/TP lead to poor TN and TP removal performance, and additional carbon source and chemical phosphorus removal agents were provided to enhance TN and TP removal in the CAST-process WWTP. A diagnostic method for the optimal operation of WWTP was established by setting 40 min of agitation before aeration. The average TN removal increased to 67.0% and showed stable TN performance after WWTP optimization. The microbial activity of Rhodocyclaceae was inhibited by the addition of chemical phosphorus removal agents. Therefore, the type of chemical phosphorus removal agents must be preliminary investigated on the basis of the dosage and TP and COD removal properties. TP removal of the CAST process reached 97% after changing the agent, and the increase of the specific TP removal to 0.05 mgTP/m, resulted in the reduction of treatment cost by 0.008 CNY/t.
- The references list is not uniform
Response: The references had been revised according to the standard of the journal WATER.
In addition, the language and format of the paper had both been revised and marked in yellow to achieve the publication of this manuscript.
We tried our best to improve the manuscript and made some changes in the manuscript. These changes will not influence the content and framework of the paper. We appreciate for your warm work earnestly, and hope that the correction will meet with approval.
Special thanks to you for your good comments.
Kind Regards
Chong LIU

Reviewer 2 Report
In this manuscript a strategy for diagnosis and optimization of activated sludge plants is presented. Although the work provides relevant data by quantifying important kinetic parameters and provide insights into potential effects from different phosphorous precipitant agent, major modifications are required before publication of these results. First, the introduction does not provide a basis / background on why and how this diagnostic (and optimization) approach was selected and/or developed. There is no reference to other similar studies (only the mention that are few studies at lines 47 and 48) or studies using the model adopted here. A second main drawback is that there is no information regarding the model adopted here, besides the name of the simulation software. Information such which model was used (e.g. ASM1 or ASM3), the parameters adopted, the influent characterization, and simulation approach (e.g. steady state simulations after model calibration with data from the batch assays) are essential for an interpretation of the results. Moreover, the manuscript structure is sometimes very difficult to follow. In some sections - as 3.1.2.1 and 3.1.2.2 – starts with an overview of nitrification and denitrification processes, which are normally found in the introduction. The last major revision aspect is regarding the effect from the different phosphorous precipitation agents to the sludge biomass. In lines 335 and 336, the authors used the term “certain impact” and later in lines 338 and 339 a “certain adverse effect” to explain changes in the microbiome driven by external agents. However, the time-line of the experiments is not clear: was the sludge analyzed just after one precipitation assay? In this case could not be that sludge was already adapted to one of the precipitation agents; and a further adaptation can also not be disclosed, hence, making the claims above very difficult to support. And finally, are other literature references providing some inhibitory range for these compounds?
Further additional remarks:
- The assays for determination of kinetic rates were performed in triplicates?
- In Figure 3 and 5 it seems also that some online sensors were used, in this is the case, please provide information regarding the type of sensors that were used.
- In Figure 6 is not clear which diagram (left/right) refers to which different biomass.
- If possible provide some reference for nitrification and denitrification rates from other investigations.
Author Response
Dear Reviewer #2:
Thank you for your letter and the comments concerning our manuscript entitled "A Diagnostic Method for Enhancing Nitrogen and Phosphorus Removal in CAST Process Wastewater Treatment Plant". (Manuscript No. Water-1753391).
The comments are all valuable and very helpful for revising and improving our paper, as well as the important guiding significance to our researches. We have studied comments carefully and have made corrections which we hope meet with approval. Revised portion are also marking with yellow in the revised paper. The main corrections in the paper and the responds to the comments of reviewer #2 are as following:
Responds to the reviewer’s comments:
In this manuscript a strategy for diagnosis and optimization of activated sludge plants is presented. Although the work provides relevant data by quantifying important kinetic parameters and provide insights into potential effects from different phosphorous precipitant agent, major modifications are required before publication of these results. First, the introduction does not provide a basis / background on why and how this diagnostic (and optimization) approach was selected and/or developed. There is no reference to other similar studies (only the mention that are few studies at lines 47 and 48) or studies using the model adopted here. A second main drawback is that there is no information regarding the model adopted here, besides the name of the simulation software. Information such which model was used (e.g. ASM1 or ASM3), the parameters adopted, the influent characterization, and simulation approach (e.g. steady state simulations after model calibration with data from the batch assays) are essential for an interpretation of the results. Moreover, the manuscript structure is sometimes very difficult to follow. In some sections - as 3.1.2.1 and 3.1.2.2 – starts with an overview of nitrification and denitrification processes, which are normally found in the introduction. The last major revision aspect is regarding the effect from the different phosphorous precipitation agents to the sludge biomass. In lines 335 and 336, the authors used the term “certain impact” and later in lines 338 and 339 a “certain adverse effect” to explain changes in the microbiome driven by external agents. However, the time-line of the experiments is not clear: was the sludge analyzed just after one precipitation assay? In this case could not be that sludge was already adapted to one of the precipitation agents; and a further adaptation can also not be disclosed, hence, making the claims above very difficult to support. And finally, are other literature references providing some inhibitory range for these compounds?
Response: The diagnostic method had been added in Introduction section, and model adopted in BioWin is ASM2, the parameters employed were provided in section 3.1.4.1 and 3.1.4.2. In the introduction, the authors mainly introduces the importance and innovation of using diagnostic methods for WWTPs. In order to ensure the logic and coherence of the manuscript, the authors believe that the description of the characteristics of the nitrification process and denitrification process is placed in 3.1.2.1 and 3.1.2.2, which is more conducive to the expression of important information. In addition, the characteristics of the nitrification process and denitrification process had been carefully revised accordingly. Moreover, many literatures have proved that long-term dosing of chemical phosphorus removal agents will have an negative impact on the bioactivity of PAOs. Therefore, the analysis of the influence of chemical phosphorus removal in this study is the result of long-term research, not the results obtained after one-time dosing of chemical phosphorus removal agents. Furthermore, literature studies have found that the dosage of chemical phosphorus removal agents has different effects on the phosphorus removal of activated sludge. The main reason is that the influent TP concentration is different and the influent TP composition is also different.
- The assays for determination of kinetic rates were performed in triplicates?
Response: Yes, the nitrification rate, denitrification rate and phosphorus release rate had been performed in triplicates.
- In Figure 3 and 5 it seems also that some online sensors were used, in this is the case, please provide information regarding the type of sensors that were used.
Response: The data of TN and ammonia nitrogen are obtained and determined manually, which is common in China.
- In Figure 6 is not clear which diagram (left/right) refers to which different biomass.
Response: The figure title had been revised.
- If possible provide some reference for nitrification and denitrification rates from other investigations.
Response: The statistics on nitrification and denitrification rates have been added in the figure title of Figure 7 and Figure 8a.
In addition, the language and format of the paper had both been revised and marked in yellow to achieve the publication of this manuscript.
We tried our best to improve the manuscript and made some changes in the manuscript. These changes will not influence the content and framework of the paper. We appreciate for your warm work earnestly, and hope that the correction will meet with approval.
Special thanks to you for your good comments.
Kind Regards
Chong LIU

Reviewer 3 Report
This manuscript presents a diagnostic method for the optimal operation of a cyclic activated sludge technology (CAST) process wastewater treatment plant in China for enhancing nitrogen and phosphorus removal. According to this study, additional carbon sources and chemical phosphorus removal agents enhance TN and TP removal in the CAST process. The work presented in this paper is interesting, original, and could contribute to the optimal performance of the investigated WWTPs. Thus, the topic is of broad interest and may fall within the scope of the journal. In order to improve the quality of this paper, I made a list of suggestions that should be considered. A major revision is required.
Title of the paper:
Line 3: To avoid abbreviations in the title, I recommend using CAST's full name.
Material and methods:
Line 72: A description of the Biowin model should be provided for more clarity for the reader.
The methodology to determine the microbial community structure is missing in this section.
Results and discussion
Line 152: Ovoid Starting paragraph with figures, put the text, then figures or tables.
There is a discrepancy between the TN values you mention in the text and those shown in Figure 3; you mention the highest value of 62 mg/l in the text; however, when I’ve checked Figure 3, I discovered that TN did not exceed 50 mg/l during the entire experiment period; all of the text must be revised accordingly.
Lines 164-169: same issue; a discrepancy between the influent ammonia content values you mention in the text and those shown in Figure 5; must be revised.
What are the discharge standards in your country? They should be indicated in a table.
Lines 163 + 172: Figure 5 is repeated twice throughout the paper. Revise figures' numbering throughout the whole paper.
Line 196: Figure 6 or 7 ?????
Lines 180-184: There is a discordance between the values of TP you mention in the text and those indicated in Figure 9; you mention TP's highest value of 26 mg/l in influent in the text; however, when I checked Figure 9, I found that TP did not exceed 16 mg/l during the whole experiment period. The same problem for the lowest value; all the text must be revised accordingly.
Author Response
Dear Reviewer #3:
Thank you for your letter and the comments concerning our manuscript entitled "A Diagnostic Method for Enhancing Nitrogen and Phosphorus Removal in CAST Process Wastewater Treatment Plant". (Manuscript No. Water-1753391).
The comments are all valuable and very helpful for revising and improving our paper, as well as the important guiding significance to our researches. We have studied comments carefully and have made corrections which we hope meet with approval. Revised portion are also marking with yellow in the revised paper. The main corrections in the paper and the responds to the comments of reviewer #3 are as following:
Responds to the reviewer’s comments:
This manuscript presents a diagnostic method for the optimal operation of a cyclic activated sludge technology (CAST) process wastewater treatment plant in China for enhancing nitrogen and phosphorus removal. According to this study, additional carbon sources and chemical phosphorus removal agents enhance TN and TP removal in the CAST process. The work presented in this paper is interesting, original, and could contribute to the optimal performance of the investigated WWTPs. Thus, the topic is of broad interest and may fall within the scope of the journal. In order to improve the quality of this paper, I made a list of suggestions that should be considered. A major revision is required.
- Line 3: To avoid abbreviations in the title, I recommend using CAST's full name.
Response: The title had been revised as “Diagnostic Method for Enhancing Nitrogen and Phosphorus Removal in Cyclic Activated Sludge Technology (CAST) process Wastewater Treatment Plant”.
- Line 72: A description of the Biowin model should be provided for more clarity for the reader.
Response: The model adopted in BioWin is ASM2, the parameters employed were provided in section 3.1.4.1 and 3.1.4.2.
- The methodology to determine the microbial community structure is missing in this section.
Response: The method for analyzing microbial community structure had been added in section 2.2.2.
- Line 152: Ovoid Starting paragraph with figures, put the text, then figures or tables.
Response: The formatting errors had been carefully revised.
5-10. There is a discrepancy between the TN values you mention in the text and those shown in Figure 3; you mention the highest value of 62 mg/l in the text; however, when I’ve checked Figure 3, I discovered that TN did not exceed 50 mg/l during the entire experiment period; all of the text must be revised accordingly. Lines 164-169: same issue; a discrepancy between the influent ammonia content values you mention in the text and those shown in Figure 5; must be revised. What are the discharge standards in your country? They should be indicated in a table. Lines 163 + 172: Figure 5 is repeated twice throughout the paper. Revise figures' numbering throughout the whole paper. Line 196: Figure 6 or 7 ????? Lines 180-184: There is a discordance between the values of TP you mention in the text and those indicated in Figure 9; you mention TP's highest value of 26 mg/l in influent in the text; however, when I checked Figure 9, I found that TP did not exceed 16 mg/l during the whole experiment period. The same problem for the lowest value; all the text must be revised accordingly.
Response: The data, figures resolution and figures numbering had all been carefully revised. In addition, the discharge standard of ammonia nitrogen, TN and TP were added in the manuscript because there are many figures and tables.
In addition, the language and format of the paper had both been revised and marked in yellow to achieve the publication of this manuscript.
We tried our best to improve the manuscript and made some changes in the manuscript. These changes will not influence the content and framework of the paper. We appreciate for your warm work earnestly, and hope that the correction will meet with approval.
Special thanks to you for your good comments.
Kind Regards
Chong LIU

Reviewer 4 Report
1- Revise the English language of the manuscript
2- What is the Composition of WWTPs Processes (example: conventional activated sludge process, anaerobic-anoxic-oxic etc …)
3- Specify in a table the amount of nitrogen, phosphorous and carbon before and after enhancing the removal process (influent characteristics)
4- Is there any effect of pH and temperature on nitrogen and phosphorous removal
Author Response
Dear Reviewer #4:
Thank you for your letter and the comments concerning our manuscript entitled "A Diagnostic Method for Enhancing Nitrogen and Phosphorus Removal in CAST Process Wastewater Treatment Plant". (Manuscript No. Water-1753391).
The comments are all valuable and very helpful for revising and improving our paper, as well as the important guiding significance to our researches. We have studied comments carefully and have made corrections which we hope meet with approval. Revised portion are also marking with yellow in the revised paper. The main corrections in the paper and the responds to the comments of reviewer #4 are as following:
Responds to the reviewer’s comments:
- Revise the English language of the manuscript
Response: The grammatical and formatting errors had been carefully revised, and the “CERTIFICATE OF ENGLISH EDITING” had been uploaded to the submission system.
- What is the Composition of WWTPs Processes (example: conventional activated sludge process, anaerobic-anoxic-oxic etc …)
Response: The composition of WWTPs processes is cyclic activated sludge technology (CAST) process.
- Specify in a table the amount of nitrogen, phosphorous and carbon before and after enhancing the removal process (influent characteristics)
Response: The effluent of ammonia nitrogen, TN and TP after optimization were added in the manuscript because there are many figures and tables.
- Is there any effect of pH and temperature on nitrogen and phosphorous removal
Response: Changes in pH and temperature will have impact on the removal of nitrogen and phosphorus, but slight fluctuations in pH and temperature during the operation of the WWTPs will not have great impact on the removal of nitrogen and phosphorus. In the future, the authors will pay more attention to the effect of pH and temperature on the removal of pollutants, so as to ensure that the proposed diagnostic method is more thorough and reasonable.
In addition, the language and format of the paper had both been revised and marked in yellow to achieve the publication of this manuscript.
We tried our best to improve the manuscript and made some changes in the manuscript. These changes will not influence the content and framework of the paper. We appreciate for your warm work earnestly, and hope that the correction will meet with approval.
Special thanks to you for your good comments.
Kind Regards
Chong LIU

Round 2
Reviewer 3 Report
Please check carefully your paper carfully and correct the mistakes in figures's numbering:
Line 180: Figure 6
Line 204: Figure 7 ; indicates a and b on the corresponding figure
Line 224:: Figure 8a
Line : 259, Figure10
Line 291: Figure 12
Line 289: Stat the paragraph 3.2.1 by text then put figures
Author Response
The manuscript had been carefully revised according to the comments.
Reviewer 4 Report
The manuscript can be accepted in its present form
Author Response
Thanks for your comments.